psychology

pandemic, COVID-19, exponential growth, linear scaling, logarithmic scaling, contextual framing

**Author for correspondence:**
Florian Hutzler
e-mail: florian.hutzler@sbg.ac.at

# Anticipating trajectories of exponential growth

Florian Hutzler, Fabio Richlan, Michael Christian Leitner, Sarah Schuster, Mario Braun and Stefan Hawelka

Department of Psychology, Centre for Cognitive Neuroscience, Paris-Lodron-University of Salzburg, Hellbrunnerstrasse 34, 5020 Salzburg, Austria

FH, 0000-0001-8195-4911; FR, 0000-0001-5373-3425;
MCL, 0000-0002-9341-854X; SS, 0000-0002-1905-4450;
MB, 0000-0001-5054-4960; SH, 0000-0002-5304-8521

Humans grossly underestimate exponential growth, but are at the same time overconfident in their (poor) judgement. The so-called 'exponential growth bias' is of new relevance in the context of COVID-19, because it explains why humans have fundamental difficulties to grasp the magnitude of a spreading epidemic. Here, we addressed the question, whether logarithmic scaling and contextual framing of epidemiological data affect the anticipation of exponential growth. Our findings show that underestimations were most pronounced when growth curves were linearly scaled *and* framed in the context of a more advanced epidemic progression. For logarithmic scaling, estimates were much more accurate, on target for growth rates around 31%, and not affected by contextual framing. We conclude that the logarithmic depiction is conducive for detecting exponential growth during an early phase as well as resurgences of exponential growth.

## 1. Anticipating trajectories of exponential growth

Humans are woefully inept at intuitively grasping exponential growth functions. Laypersons as well as political decision makers grossly underestimate exponential growth [1], regardless of being presented with numerical, graphical or non-quantitative representations [2]. At the same time, however, humans are overconfident despite their poor judgements [3]. This effect was hitherto primarily investigated in the domain of economics concerning, for example, the anticipation of the development of interests, and called the 'exponential growth bias' [4].

Although from a slightly different perspective, this effect is of new relevance with respect to the coronavirus disease 2019 (COVID-19) [5]. The inability to anticipate exponential growth left many, the public [6] as well as political decision makers,

**Figure 1.** Data from the COVID-19 epidemic: (*a*) illustrates that an early phase of exponential growth (UK) could elude the human beholder when being linearly scaled and framed in the context of a more advanced epidemic (Italy), whereas an appropriate range in (*b*) clearly reveals its exponential nature. Logarithmic scaling in (*c*) might allow observers to linearly extrapolate and hence detect the presence of exponential growth.

unprepared for the rapid increase in COVID-19 cases. Here, we hypothesize that the way in which epidemic data are visualized might be a crucial factor for attenuating the exponential growth bias. First, we hypothesize that logarithmic scaling will facilitate anticipating the sheer magnitude of exponential growth. Logarithmic scaling might allow the beholder to apply linear visual extrapolation, because here exponential functions are plotted as straight lines. While being the obvious choice from a mathematical perspective, little is known about how logarithmic plotting affects human anticipation of exponential growth. Logarithmic plotting was not considered as conducive for the accurate anticipation of interest rates in economics, presumably because logarithmic compression trades awareness of exponential growth for attenuated accuracy. However, concerning an arising epidemic, the mere detection of exponential growth *per se* is of importance, but not so much the exact estimate of the final numbers.

Second, we hypothesize that linear scaling of exponential growth is especially susceptible to *framing effects* [7] when we compare emerging outbreaks to regions with a more advanced progression of the epidemic. To illustrate, when trying to anticipate the development of COVID-19 in their respective countries, people compared the number of cases to that of Italy—Europe's first major COVID-19 outbreak. Examining the number of confirmed cases in Italy in 2020 as depicted in figure 1 [8] an observer will readily identify exponential growth. However, humans might not detect a relatively early phase of exponential growth, as e.g. in the United Kingdom during that time, when it is

framed[1] in the context of the more advanced outbreak. However, when we adapt the scale to the range of the data of the UK—as in figure 1b—an observer can spot exponential growth also for the UK and note that—although the absolute numbers differ—the trajectories are similar. The similarity of the trajectories is even more evident when the data are plotted with a logarithmic scale as depicted in figure 1c.

To investigate these hypotheses, we presented participants with graphs illustrating the first 20 days of the spread of a hypothetical virus and asked them for their best, intuitive estimate of the number of cases for the 30th day. During the initial 20 days, this hypothetical outbreak is, similar to the actual progression of COVID-19, exhibiting exponential growth (model estimates of [9]). The subsequent period of 10 days can be assumed to be largely unaffected by potential interventions such as social distancing [10]. We orthogonally varied the *framing* and *type of scale*, that is, the data was either *framed* in the context of a more advanced epidemic (Y-axis ranging to 10 000) or *unframed*, that is, appropriately scaled for the range of the data at the 20th day (Y-axis ranging to 1000). Concerning *type of scale*, the Y-axis was either linearly scaled or logarithmically scaled (see Method section for exemplary stimuli). For each of the four conditions, the exponential growth of five different, hypothetical viruses was simulated by using mean growth rates between 31 and 38% and a continuous growth rate reduction to mimic the data from the COVID-19 epidemic experienced by the participants (see Material and methods for details). The growth functions accumulate to target values of 2000, 4000, 6000, 8000 and 10 000 infections on day 30, respectively.

# 2. Material and methods

## 2.1. Participants

A 122 volunteers participated in the experiment (age range: 18–74 years, mean = 30 years, s.d. = 13.3, $n$ = 33 male, educational attainment: bachelor degree or higher: 43%, high school: 36%, other: 11%). Participants were recruited by snowball sampling using social media.

## 2.2. Stimuli

Exemplary stimuli are provided in figure 2. We simulated the spreading of five hypothetical viruses using exponential growth functions. All growth functions started with a growth rate of 100% in order to realize a whole number first increment from the first to the second day. In order to mimic the growth rates experienced by the participants during the course of the COVID-19 epidemic, growth rates were continuously reduced using 30-day reduction vectors. Mean and final growth rates for the five functions were $M$ = 31% (final growth rate: 22%), $M$ = 34% (final: 25%), $M$ = 36% (final: 27%), $M$ = 37% (final: 28%) and $M$ = 38% (final 29%).

Mean growth rates were around and above that of the United States during March 2020 (34.5%; Dong *et al.* [8]). Growth rate reduction in COVID-19 cases, which we mimicked in the present study, facilitates the estimation for the participants, since although people tend to extrapolate exponentially, they do so with too small an exponent [2]. Figure 3 depicts the course of the five hypothetical virus spreads amounting to target values of 2000, 4000, 6000, 8000 and 10 000 cases on day 30, respectively.

## 2.3. Procedure

All data were acquired online between the 11 and 24 April 2020 with the survey tool *LimeSurvey* version 3.22.2+200204 (Hamburg, Germany). In the introductory phase, participants were familiarized with the types of figures used in the experiments, briefed on logarithmic scales and were explicitly made aware of the different types of scales (logarithmic and linear) as well as the different ranges of the Y-axes (range 1000 and 10 000, respectively). Participants were instructed to give their best intuitive estimate of the number of cases at day 30 and to enter their estimate numerically in a field below the figure, and they were informed that the time available for each figure was 2 min. Subsequently, the 20 experimental stimuli were presented one after another, that is, on an individual page. They were visible until the

---

[1]In order to acknowledge Kahneman's [7] *framing effect*, the terms *framed* and *unframed* are used throughout the manuscript to indicate whether or not data are presented in the context of a more advanced outbreak or in a range that is appropriate for the data, respectively.

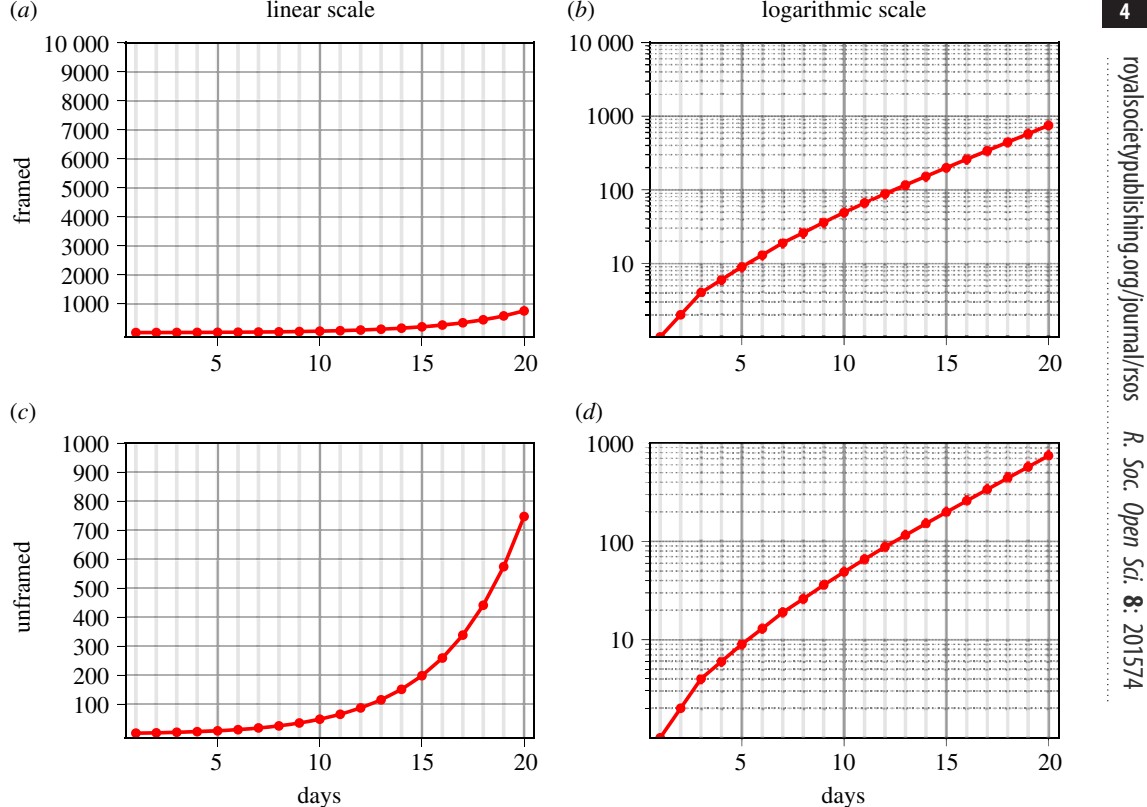

**Figure 2.** Exemplary experimental stimuli. The panels depict the spread of a hypothetical virus from the 1st to the 20th day while participants had to estimate the number of cases on day 30. Exponential growth was either framed in the context of a more advanced epidemic (*a,b*) or scaled to the range of the data on the 20th day (*c,d*) and they were either plotted using a linear scale (*a,c*) or a logarithmic scale (*b,d*).

participant entered his/her response or until the time limit expired. The order of presentation was randomized for every participant. At the end of the experiment, the demographic data of the participants were queried.

## 2.4. Analysis

In total, our study would have yielded 2440 responses (122 participants x 20 figures), but 15 responses (0.61% of the data) were missing, because the response time limit expired. Of the remaining responses, 27 (1.11%) were removed, because they exceeded the cut-off criterion of 50 000 (i.e. five times the maximum target value). Subsequently, we removed (individually for each experimental condition) 22 responses (0.90% of the data) which exceeded the mean of their respective condition by 5 s.d. Two participants for whom this procedure resulted in less than 15 out of 20 responses were removed from the analysis. In sum, 2360 responses (96.72% of the data) entered the analysis. To facilitate the interpretation, we transformed the estimates of the participants into the percentage of the actual target value (e.g. for the target value of 10 000 an estimate of 7000 is 70%).

## 3. Results

We first provide a visual-verbal description of our findings, before we present the statistical details. Figure 4 reveals that—even at times when people are highly aware of the intricacy of exponential growth due to the ongoing COVID-19 epidemic—they nevertheless underestimate it fundamentally. It is evident from the figure that the underestimations are, as we hypothesized, most pronounced when the data were linearly scaled and framed in the context of a more advanced epidemic progression. By contrast, predictions for logarithmic scaling were near the target for growth rates of 31% and better than those for linear scaling for a growth rate of 34%.

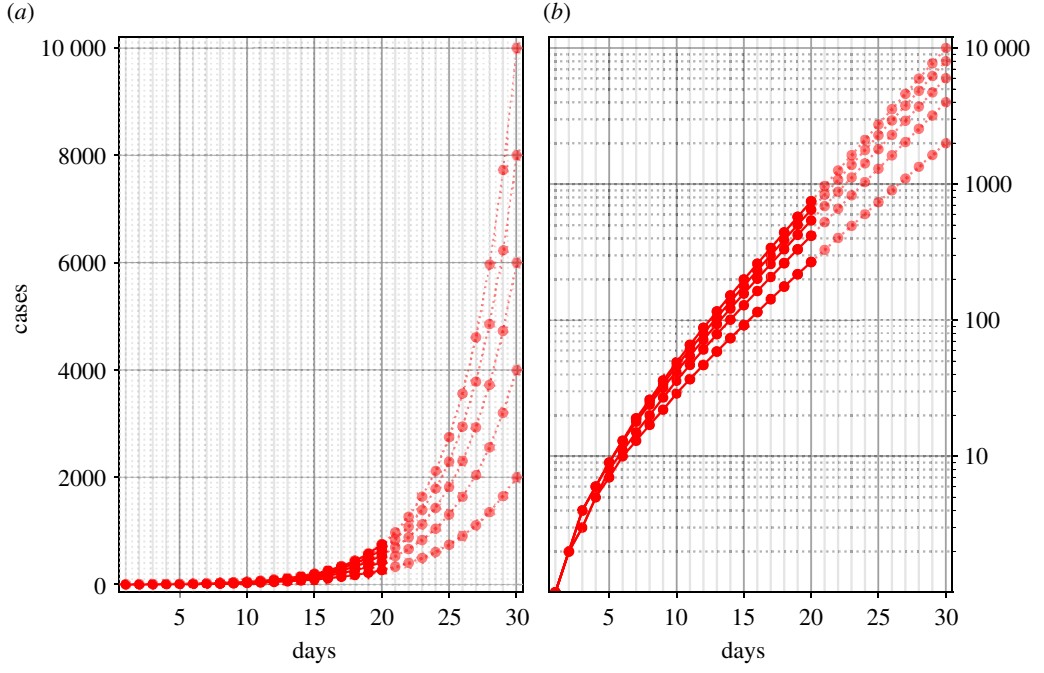

**Figure 3.** Growth rates and target value at day 30. The figure shows the five different lines which the participants saw up to day 20 (solid lines) and their continuation towards day 30 (transparent lines) in linear and logarithmic scale.

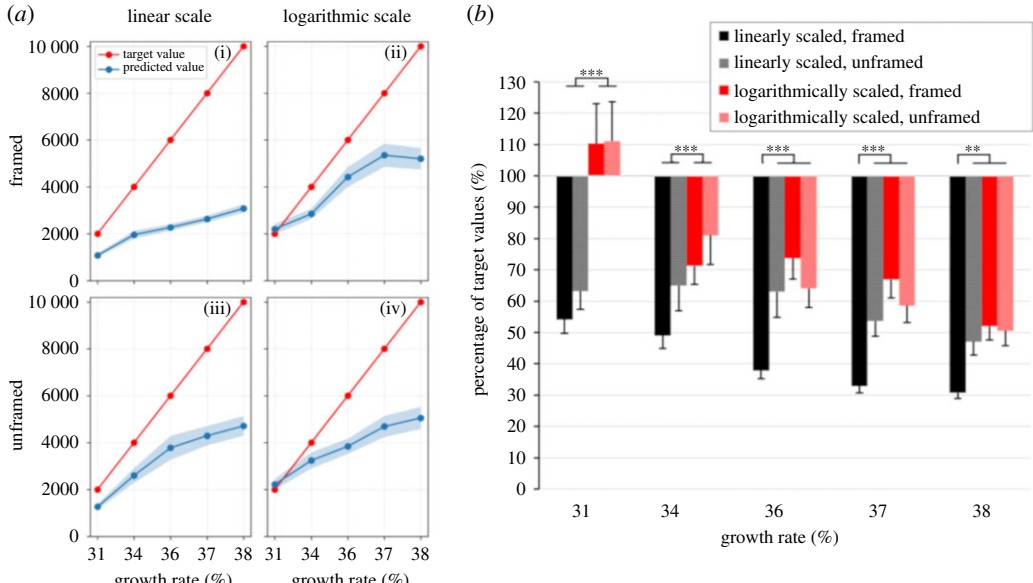

**Figure 4.** Underestimation of exponential growth: (*a*) shows the participants' prediction of the number of cases on day 30 compared to the correct target values for different growth rates—depicted for the four experimental conditions in separate panels; (*b*) shows the predictions transformed into percentages of the target value. The asterisks indicate significant differences between conditions (**$p < 0.01$, ***$p < 0.001$) estimated by pairwise comparisons.

For statistical analysis, we submitted the percentages of the target values to a $5 \times 2 \times 2$ repeated-measures ANOVA with *growth rate* (31%, 34%, 36%, 37% and 38%), *framing* (framed versus unframed) and *type of scale* (linear versus logarithmic) as within-subject factors. An interaction of *type of scale* by *framing*, $F_{1,93} = 18.25$, $p < 0.001$, and an interaction of *type of scale* by *growth rate*, $F_{4,372} = 10.45$, $p < 0.001$, qualified the main effects of *type of scale*, $F_{1,93} = 25.75$, $p < 0.001$, and *growth rate*, $F_{4,372} = 27.75$, $p < 0.001$. Neither the main effect of *framing*, $F_{1,93} = 2.69$, $p = 0.10$, nor the interaction of *framing* by *growth rate*, $F < 1$, or the triple interaction, $F_{4,372} = 1.97$, $p = 0.10$, were significant.

We conducted *post hoc* tests separately for the five different growth rates using $2 \times 2$ repeated-measures ANOVAs with *framing* and *type of scale* as within-subject factors. For the two smallest growth rates of 31% and 34%, main effects of *type of scale* indicated significantly larger underestimations for linearly scaled figures than for logarithmically scaled figures, 31%: $F_{1,117} = 29.77$, $p < 0.001$ and 34%: $F_{1,112} = 12.23$, $p < 0.001$. Neither the main effects of *framing* nor the interactions were significant, all $F$s less than 1.02. For the growth rates of 36%, 37% and 38%, an interaction of *type of scale* by *framing* qualified the main effects, $F_{1,112} = 9.54$, $p < 0.01$, $F_{1,111} = 14.39$, $p < 0.001$, and, of borderline significance, $F_{1,106} = 3.53$, $p = 0.06$. Additional *post hoc* tests revealed significantly larger underestimations for linearly scaled, framed figures compared to the remaining three conditions, all $F$s > 7.25, all $p$s < 0.01. Predictions for the remaining three conditions did not differ from each other in all three conditions, all $F$s less than 2.8 (except for a stronger underestimation for linearly scaled, unframed figures compared to logarithmically scaled, framed figures for a growth rate of 37%, $F_{1,115} = 4.09$, $p < 0.05$).

## 4. Discussion

The objective of the present study was to assess whether the way in which epidemic data is visualized might help to attenuate the exponential growth bias. Our findings suggest that a linear visual extrapolation—made possible by the logarithmic scaling of exponential growth—allows the beholder a more intuitive anticipation of exponential growth at least for growth rates of around 31% which are quite typical for the present COVID-19 epidemic. Moreover, with logarithmic scaling, the participants' predictions were not affected by the differences in framing, that is, the range of the *y*-axis. Thus, logarithmic scaling makes it possible to visualize epidemic trajectories with different progressions (e.g. early versus advanced epidemic growth) in one single graph. Put differently, visualizing exponential growth using logarithmic scales is particularly suited in the early phase of an epidemic, because it reveals the similarity in the trajectory with a more advanced epidemic growth which otherwise would elude the observer. Our findings are also relevant for resurgences of exponential growth of COVID-19 cases which need to be conveyed—ideally—as early as possible. To allow beholders to intuitively grasp the development of trajectories will be essential for public to embrace policies such as mask wearing or stay-at-home orders, but also for politicians and authorities hesitant to issue such unpopular policies. Finally, an intuitive appreciation for exponential growth might affect politicians' decisions directly, since we cannot expect them to rely on scientific advice if it contradicts their gut feeling. The present data showed that logarithmically scaled figures are superior to linear-scaled figures with respect to that.

Data accessibility. The survey, the raw data and the syntax used for analysis are available at the Open Science Framework at osf.io/9qr6y. An English translation of the survey is provided in the electronic supplementary material [11].
Authors' contributions. F.H. and S.H. developed the study concept and all authors contributed to the study design. F.R. collected the data. S.H. and F.H. analysed the data. S.H. and M.C.L. made the figures. F.H. and S.H. wrote the manuscript, and M.C.L., M.B. and S.S. provided critical revisions. All authors reviewed the manuscript.
Competing interests. We declare we have no competing interests.
Funding. This study was funded by Austrian Science Fund (P 31299).

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
