## [Peer Review File · Royal Society Open Science]

Review History

RSOS-201574.R0 (Original submission)

Review form: Reviewer 1 (Mateusz Hohol)

Is the manuscript scientifically sound in its present form?

No

Are the interpretations and conclusions justified by the results?

Yes

Is the language acceptable?

Yes

Do you have any ethical concerns with this paper?

No

Have you any concerns about statistical analyses in this paper?

No

Recommendation?

Accept with minor revision (please list in comments)

Comments to the Author(s)

In the manuscript, the authors address a topic of high contemporary importance - perception biases related to the reading plots depicting exponential growth of phenomena, esp. infections in pandemic scenarios.

The manuscript is well-written and easy to follow. I have no concerns about statistical analyses and conclusions are sound. Given the enormous importance of relevant communication of scientific data in the current pandemic circumstances, it would be most probably of high interest to the scientific community in general. Moreover, the presented study could be interesting for specialists, e.g., cognitive psychologists interested in biases and numerical cognition researchers. This being said, there are few minor blemishes that should be addressed before the final acceptance of the manuscript:

-I suggest moving "Material and methods" section from the supplementary materials to the main manuscript.

-More extensive demographics (e.g., level of education) of the research sample and recruitment procedure should be provided.

-The authors should provide a time frame when the research was conducted (throughout the year, the exposition of the participants to exponential plots via consumption of the media probably steadily increased; thus, it might contribute to the overall ability to read them properly).

-It is unclear whether the study was conducted entirely online or "stationary."

-Pandemic-related literature on exponential growth bias could be more up to date (please check e.g., Lammers, Crusius, Gast, 2020, PNAS; Banerjee, Bhattacharya, Majumdar, 2021, Social Science & Medicine).

-p4 l38 "...logarithmic compression trades an increased chance for an awareness of exponential growth for attenuated accuracy" - the reference should be provided.

-p11 l10 "M=30 years" it is unclear whether the "M" symbol refers to the mean or the median, standard deviation (SD) should also be provided.

-p11 l12 "33 male" it is unclear whether the number describes n of males in the entire participant group or e.g. percentage of males.

-The raw data and syntax used for analysis are available at the OSF and exemplary stimuli are presented in the manuscript. However, to ensure replicability and follow-ups, the survey (along with stimuli graphs) used in the study should be shared at the OSF as well.

Review form: Reviewer 2 (Joshua Tasoff)**Is the manuscript scientifically sound in its present form?**

Yes

Are the interpretations and conclusions justified by the results?

No

Is the language acceptable?

Yes

Do you have any ethical concerns with this paper?

No

Have you any concerns about statistical analyses in this paper?

No

Recommendation?

Major revision is needed (please make suggestions in comments)

Comments to the Author(s)

See attachment (Appendix A).

Decision letter (RSOS-201574.R0)

Dear Dr Hawelka

The Editors assigned to your paper RSOS-201574 "Anticipating trajectories of exponential growth" have now received comments from reviewers and would like you to revise the paper in accordance with the reviewer comments and any comments from the Editors. Please note this decision does not guarantee eventual acceptance.

Please submit your revised manuscript and required files (see below) no later than 21 days from today's (ie 09-Feb-2021) date. Note: the ScholarOne system will 'lock' if submission of the revision is attempted 21 or more days after the deadline. If you do not think you will be able to meet this deadline please contact the editorial office immediately.

on behalf of Dr Denes Szucs (Associate Editor) and Essi Viding (Subject Editor)
 openscience@royalsociety.org

Reviewer comments to Author:

Reviewer: 1

Comments to the Author(s)

In the manuscript, the authors address a topic of high contemporary importance - perception biases related to the reading plots depicting exponential growth of phenomena, esp. infections in pandemic scenarios.

The manuscript is well-written and easy to follow. I have no concerns about statistical analyses and conclusions are sound. Given the enormous importance of relevant communication of scientific data in the current pandemic circumstances, it would be most probably of high interest to the scientific community in general. Moreover, the presented study could be interesting for specialists, e.g., cognitive psychologists interested in biases and numerical cognition researchers. This being said, there are few minor blemishes that should be addressed before the final acceptance of the manuscript:

- I suggest moving “Material and methods” section from the supplementary materials to the main manuscript.
- More extensive demographics (e.g., level of education) of the research sample and recruitment procedure should be provided.
- The authors should provide a time frame when the research was conducted (throughout the year, the exposition of the participants to exponential plots via consumption of the media probably steadily increased; thus, it might contribute to the overall ability to read them properly).
- It is unclear whether the study was conducted entirely online or “stationary.”
- Pandemic-related literature on exponential growth bias could be more up to date (please check e.g., Lammers, Crusius, Gast, 2020, PNAS; Banerjee, Bhattacharya, Majumdar, 2021, Social Science & Medicine).
- p4 l38 “...logarithmic compression trades an increased chance for an awareness of exponential growth for attenuated accuracy” - the reference should be provided.
- p11 l10 “M=30 years” it is unclear whether the “M” symbol refers to the mean or the median, standard deviation (SD) should also be provided.
- p11 l12 “33 male” it is unclear whether the number describes n of males in the entire participant group or e.g. percentage of males.
- The raw data and syntax used for analysis are available at the OSF and exemplary stimuli are presented in the manuscript. However, to ensure replicability and follow-ups, the survey (along with stimuli graphs) used in the study should be shared at the OSF as well.

Reviewer: 2

Comments to the Author(s)

See attachment

===PREPARING YOUR MANUSCRIPT===

one version identifying all the changes that have been made (for instance, in coloured highlight, in bold text, or tracked changes);
a 'clean' version of the new manuscript that incorporates the changes made, but does not highlight them. This version will be used for typesetting if your manuscript is accepted. Please ensure that any equations included in the paper are editable text and not embedded images.

===PREPARING YOUR REVISION IN SCHOLARONE===

- Any electronic supplementary material (ESM).
- If you are requesting a discretionary waiver for the article processing charge, the waiver form must be included at this step.
- If you are providing image files for potential cover images, please upload these at this step, and inform the editorial office you have done so. You must hold the copyright to any image provided.
- A copy of your point-by-point response to referees and Editors. This will expedite the preparation of your proof.

- Ensure that your data access statement meets the requirements at <https://royalsociety.org/journals/authors/author-guidelines/#data>. You should ensure that you cite the dataset in your reference list. If you have deposited data etc in the Dryad repository, please include both the 'For publication' link and 'For review' link at this stage.
- If you are requesting an article processing charge waiver, you must select the relevant waiver option (if requesting a discretionary waiver, the form should have been uploaded at Step 3 'File upload' above).
- If you have uploaded ESM files, please ensure you follow the guidance at <https://royalsociety.org/journals/authors/author-guidelines/#supplementary-material> to include a suitable title and informative caption. An example of appropriate titling and captioning may be found at https://figshare.com/articles/Table_S2_from_Is_there_a_trade-off_between_peak_performance_and_performance_breadth_across_temperatures_for_aerobic_scorpions_in_teleost_fishes_/3843624.

Author's Response to Decision Letter for (RSOS-201574.R0)

See Appendix B.

RSOS-201574.R1 (Revision)

Review form: Reviewer 1 (Mateusz Hohol)

Is the manuscript scientifically sound in its present form?

Yes

Are the interpretations and conclusions justified by the results?

Yes

Is the language acceptable?

Yes

Do you have any ethical concerns with this paper?

No

Have you any concerns about statistical analyses in this paper?

No

Recommendation?

Accept as is

Comments to the Author(s)

Thank you for incorporating all the suggestions raised in the previous round. In my opinion, the manuscript is ready for publishing. Thank you for your valuable and interesting contribution. All the best, Mateusz Hohol

Review form: Reviewer 2 (Joshua Tasoff)**Is the manuscript scientifically sound in its present form?**

Yes

Are the interpretations and conclusions justified by the results?

Yes

Is the language acceptable?

Yes

Do you have any ethical concerns with this paper?

No

Have you any concerns about statistical analyses in this paper?

No

Recommendation?

Accept with minor revision (please list in comments)

Comments to the Author(s)

Please see attachment (Appendix C).

Decision letter (RSOS-201574.R1)

Dear Dr Hawelka

On behalf of the Editors, we are pleased to inform you that your Manuscript RSOS-201574.R1 "Anticipating trajectories of exponential growth" has been accepted for publication in Royal Society Open Science subject to minor revision in accordance with the referees' reports. Please find the referees' comments along with any feedback from the Editors below my signature.

Please submit your revised manuscript and required files (see below) no later than 7 days from today's (ie 29-Mar-2021) date. Note: the ScholarOne system will 'lock' if submission of the revision is attempted 7 or more days after the deadline. If you do not think you will be able to meet this deadline please contact the editorial office immediately.

on behalf of Dr Denes Szucs (Associate Editor) and Essi Viding (Subject Editor)
openscience@royalsociety.org

Associate Editor Comments to Author (Dr Denes Szucs):

Please include your survey in supplementary material as recommended by R2.

Reviewer comments to Author:

Reviewer: 1
Comments to the Author(s)

Thank you for incorporating all the suggestions raised in the previous round. In my opinion, the manuscript is ready for publishing. Thank you for your valuable and interesting contribution.
All the best, Mateusz Hohol

Reviewer: 2
Comments to the Author(s)

Please see attachment.

===PREPARING YOUR MANUSCRIPT===

one version identifying all the changes that have been made (for instance, in coloured highlight, in bold text, or tracked changes);
 a 'clean' version of the new manuscript that incorporates the changes made, but does not highlight them. This version will be used for typesetting.

===PREPARING YOUR REVISION IN SCHOLARONE===

Author's Response to Decision Letter for (RSOS-201574.R1)

See Appendix D.

Decision letter (RSOS-201574.R2)

Dear Dr Hawelka,

I am pleased to inform you that your manuscript entitled "Anticipating trajectories of exponential growth" is now accepted for publication in Royal Society Open Science.

If you have not already done so, please remember to make any data sets or code libraries 'live' prior to publication, and update any links as needed when you receive a proof to check - for

instance, from a private 'for review' URL to a publicly accessible 'for publication' URL. It is good practice to also add data sets, code and other digital materials to your reference list.

You can expect to receive a proof of your article in the near future. Please contact the editorial office (openscience@royalsociety.org) and the production office (openscience_proofs@royalsociety.org) to let us know if you are likely to be away from e-mail contact – if you are going to be away, please nominate a co-author (if available) to manage the proofing process, and ensure they are copied into your email to the journal. Due to rapid publication and an extremely tight schedule, if comments are not received, your paper may experience a delay in publication.

on behalf of Dr Denes Szucs (Associate Editor) and Essi Viding (Subject Editor)
openscience@royalsociety.org

Appendix A

Report on “Anticipating trajectories of exponential growth”

The authors conduct an experiment in which online participants are provided with figures of hypothetical disease cases over 20 days and had to predict how many cases there would be by day 30. There were 5 growth rates, early epidemic vs. advanced (this varied the base number of cases from either 100 or 1,000), and logistic vs. linear scale yielding $5 \times 2 \times 2 = 20$ conditions/figures. Each of the 122 participants received all 20 figures. The authors found that participants tend to underestimate on all linear based scales, and usually on logarithmic scales though much less so. Within the linear based scale, more advanced epidemics caused predictions to be lower.

This paper is short and to the point. The methods appear valid as far as I can tell though I request more information before making a final determination, see below. I have some comments and criticisms that I hope will improve the paper.

Comments

1. The instrument is missing and should be included. Please include in your resubmission.
2. I’m wrestling with the fact that an exponential-growth solution is not the definitively correct answer. First, even from a theoretical perspective, the number of cases should not grow exponentially, except maybe early in the process. Theoretically, my understanding is that total cases will look more like a logarithmic curve. Second, from an empirical perspective, there is no reason that a real pandemic should conform to a model-based curve (see for example Kissler et al., 2020). Real pandemics have waves. So, the presumption that people *should* be trying to do exponential forecasting is misplaced.
 - a. I suppose a response to this criticism is that the purpose of the paper is to measure forecasts relative to an exponential benchmark regardless of whether those exponential benchmarks are correct. But then that begs the question, if we know that the exponential benchmark is wrong and people don’t forecast exponentially, what then is the point being made by this paper?
3. I find the label of the treatment “framed” vs. “unframed” not descriptive of what you’re doing and therefore easily confusable. Why don’t you call them “advanced” vs. “early”, or something intuitive like this.
4. There are a few sentences where the prepositions seemed a bit off to me (but I’m American so maybe British usage is different).
 - a. On the p.3 there are a few sentences that state, “the exponential growth bias”. It sounds better to omit “the”.

- b. P. 3 I recommend changing, “trades an increased chance for an awareness of exponential growth for attenuated accuracy” to “trades awareness of exponential growth for attenuated accuracy”.

Kissler, Stephen M., Christine Tedijanto, Edward Goldstein, Yonatan H. Grad, and Marc Lipsitch. "Projecting the transmission dynamics of SARS-CoV-2 through the postpandemic period." *Science* 368, no. 6493 (2020): 860-868.

Appendix B

Response to Reviewer

Reviewer: #1

Comments to the Author(s)

In the manuscript, the authors address a topic of high contemporary importance - perception biases related to the reading plots depicting exponential growth of phenomena, esp. infections in pandemic scenarios.

The manuscript is well-written and easy to follow. I have no concerns about statistical analyses and conclusions are sound. Given the enormous importance of relevant communication of scientific data in the current pandemic circumstances, it would be most probably of high interest to the scientific community in general. Moreover, the presented study could be interesting for specialists, e.g., cognitive psychologists interested in biases and numerical cognition researchers. This being said, there are few minor blemishes that should be addressed before the final acceptance of the manuscript:

Issue 1: I suggest moving “Material and methods” section from the supplementary materials to the main manuscript.

Response: We integrated the “Material and methods” section in the Method Section of the main manuscript as suggested.

Issue 2: More extensive demographics (e.g., level of education) of the research sample and recruitment procedure should be provided.

Response: We now provide information on the level of education as well as the recruitment procedure in the participant section:

“Participants. 122 volunteers participated in the experiment (age range: 18-74 years, $M=30$ years $SD=13.3$, $n=33$ male, educational attainment: bachelor degree or higher: 43%, high school: 36%, other: 11%). Participants were recruited by snowball sampling using social media.”, P. 5

Issue 3: The authors should provide a time frame when the research was conducted (throughout the year, the exposition of the participants to exponential plots via consumption of the media probably steadily increased; thus, it might contribute to the overall ability to read them properly).

Response: Thank you for pointing out this issue, we now explicitly state the period during that the data was acquired:

“All data were acquired between the 11th and 24th of April 2020 [...]”, P. 7

Issue 4: It is unclear whether the study was conducted entirely online or “stationary.”

Response: We now explicitly state that all data were acquired online with a survey tool:

“All data were acquired online between the 11th and 24th of April 2020 with the survey tool LimeSurvey version 3.22.2+200204 (Hamburg, Germany).” P. 7

Issue 5: Pandemic-related literature on exponential growth bias could be more up to date (please check e.g., Lammers, Crusius, Gast, 2020, PNAS; Banerjee, Bhattacharya, Majumdar, 2021, Social Science & Medicine).

Response: Thank you very much for pointing out this literature. We now cite Banerjee, Bhattacharya & Majumdar (2021) when we point out the relevance of the exponential growth bias in the context of COVID-19:

"Although from a slightly different perspective, this effect is of new relevance with respect to the coronavirus disease 2019 [(COVID-19)] (Banerjee, Bhattacharya & Majumdar, 2021)" P. 3

The work of Lammers, Crusius & Gast (2020) is now mentioned when we refer to lay persons' inability to anticipate exponential growth:

"The inability to anticipate exponential growth left many, the public (see, e.g., Lammers, Crusius & Gast, 2020) as well as political decision makers, unprepared for the rapid increase in COVID-19 cases." P. 3

Further, we now also refer to the work of Flaxman et al. (2020) and Kissler et al. (2020) (see also Reviewer 2).

Issue 6: p4 l38 "...logarithmic compression trades an increased chance for an awareness of exponential growth for attenuated accuracy" - the reference should be provided.

Response: We rephrased the sentence in order indicate that the notion that logarithmic plotting was considered as not conducive for anticipation of interest rates due to the resulting inaccuracy is our presumption:

"[...] presumably because logarithmic compression trades awareness of exponential growth for attenuated accuracy." P. 3

Issue 7: p11 l10 "M=30 years" it is unclear whether the "M" symbol refers to the mean or the median, standard deviation (SD) should also be provided.

Response: We now provide the standard deviation and explicitly use "Mean" when referring to the mean age.

Issue 8: p11 l12 "33 male" it is unclear whether the number describes n of males in the entire participant group or e.g. percentage of males.

Response: We changed the expression to "n = 33 males".

Issue 9: The raw data and syntax used for analysis are available at the OSF and exemplary stimuli are presented in the manuscript. However, to ensure replicability and follow-ups, the survey (along with stimuli graphs) used in the study should be shared at the OSF as well.

Response: Thank you for pointing us to this issue. The survey including the stimuli graphs is now shared at the OSF (<https://osf.io/9qr6y/files/>), both as a LimeSurvey structure and as a HTML version (see also Reviewer 2). We updated the corresponding section of the manuscript:

"**Data and materials availability:** The survey, the raw data and the syntax used for analysis are available at the Open Science Framework at osf.io/9qr6y" P.11

Reviewer 2

Report on "Anticipating trajectories of exponential growth"

The authors conduct an experiment in which online participants are provided with figures of hypothetical disease cases over 20 days and had to predict how many cases there would be by day 30. There were 5 growth rates, early epidemic vs. advanced (this varied the base number of cases from either 100 or 1,000), and logistic vs. linear scale yielding $5 \times 2 \times 2 = 20$ conditions/figures. Each of the 122 participants received all 20 figures. The authors found that participants tend to underestimate on all linear based scales, and usually on logarithmic scales though much less so. Within the linear based scale, more advanced epidemics caused predictions to be lower.

This paper is short and to the point. The methods appear valid as far as I can tell though I request more information before making a final determination, see below. I have some comments and criticisms that I hope will improve the paper.

Comments

Issue 1: The instrument is missing and should be included. Please include in your resubmission.

Response: We now provide the instrument (survey including graphs) in the data repository at the OSF (<https://osf.io/9qr6y/files/>), both as a LimeSurvey structure and as a HTML version (see also Reviewer 1). We updated the corresponding section of the manuscript:

“Data and materials availability: The survey, the raw data and the syntax used for analysis are available at the Open Science Framework at osf.io/9qr6y” P.11

Issue 2: I'm wrestling with the fact that an exponential-growth solution is not the definitively correct answer. First, even from a theoretical perspective, the number of cases should not grow exponentially, except maybe early in the process. Theoretically, my understanding is that total cases will look more like a logarithmic curve. Second, from an empirical perspective, there is no reason that a real pandemic should conform to a model-based curve (see for example Kissler et al., 2020). Real pandemics have waves. So, the presumption that people *should* be trying to do exponential forecasting is misplaced.

- a. I suppose a response to this criticism is that the purpose of the paper is to measure forecasts relative to an exponential benchmark regardless of whether those exponential benchmarks are correct. But then that begs the question, if we know that the exponential benchmark is wrong and people don't forecast exponentially, what then is the point being made by this paper?

Response: Thank you for addressing this important issue. As pointed out by the reviewer, cases will not increase exponentially throughout the whole course of an epidemic. Rather, exponential growth is prevalent especially in an early phase of the epidemic (see, e.g., the simulations provided by Flaxman et al., 2020) before interventions (such as social distancing) or immunity can take effect. For the COVID-19 pandemic, such exponential growth was observable from the beginning of February 2020 until mid-March 2020. In the present manuscript we targeted this early exponential growth in order to explore whether humans can anticipate the dynamics of an epidemic on the basis of quite low, initial case numbers. In line with that, we presented participants with a very narrow time window, more specifically, the initial 20 days. Participants were then asked to extrapolate case numbers for the imminent future, i.e., the next 10 days. The dynamic of the epidemic growth within this period of 10 days is, in case of COVID-19, still unaffected by any countermeasures such as social distancing (Kissler et al., 2020). For these (in total) 30 days we tried to mimic the actual reduction in growth rates as observed in the empirical data (see Dong, Du & Gardner, 2020) by continuously reducing the growth rate. The reviewer's issue points us to the fact, however, that we need to elaborate on this underlying reasoning in the manuscript. In the previous version of the manuscript, the fact that we refer to an early phase of exponential growth was indicated by (highlighted in bold):

“However, humans might not detect a relatively early phase of exponential growth, as e.g. in the United Kingdom during that time, when it is framed in the context of the more advanced outbreak.” P. 3

Now, we try to elaborate the underlying reasoning by also stating explicitly that (addition in bold):

“To investigate these hypotheses, we presented participants with graphs illustrating the first 20 days of the spread of a hypothetical virus and asked them for their best, intuitive estimate of the number of cases for the 30th day. During the initial 20 days this hypothetical outbreak is, similar to the actual progression of COVID-19, exhibiting exponential growth (cf. model estimates of Flaxman et al., 2020). The subsequent period of 10 days can be assumed to be largely unaffected by potential interventions such as social distancing (cf. Kissler et al., 2020).” P. 5

Issue: I find the label of the treatment “framed” vs. “unframed” not descriptive of what you're doing and therefore easily confusable. Why don't you call them “advanced” vs. “early”, or something intuitive like this.

Response: Thank you very much for pointing out that these terms are not self explanatory. We considered using alternative terms as suggested by the reviewer, however, we are under the impression that using the terms “framed” and “unframed” allows us to explicitly refer to Kahneman’s (2003) framing effect. We try to make this reasoning more clear in footnote 1:

“In order to acknowledge Kahneman’s (2003) *framing effect*, the terms *framed* and *unframed* are used throughout the manuscript to indicate, whether or not data is presented in the context of a more advanced outbreak or in a range which is appropriate for the data, respectively.” P. 3

Issue: There are a few sentences where the prepositions seemed a bit off to me (but I’m American so maybe British usage is different).

- a. On the p.3 there are a few sentences that state, “the exponential growth bias”. It sounds better to omit “the”.
- b. P. 3 I recommend changing, “trades an increased chance for an awareness of exponential growth for attenuated accuracy” to “trades awareness of exponential growth for attenuated accuracy”.

Response: Thank you very much, we now language-edited the manuscript best to our best knowledge we could and applied the changes suggested by the reviewer..

References:

- Banerjee, R., Bhattacharya, J. & Majumdar, P. (2021). Exponential-growth prediction bias and compliance with safety measures related to COVID-19. *Social Science & Medicine*, 268 (113473). doi: 10.1016/j.socscimed.2020.113473
- Flaxman, S., Mishra, S., Gandy, A. *et al.* (2020). Estimating the effects of non-pharmaceutical interventions on COVID-19 in Europe. *Nature* 584, 257–261. doi: 10.1038/s41586-020-2405-7
- Kissler, S. M., Tedijanto, C., Goldstein, E., Grad, Y. H., & Lipsitch M. (2020). Projecting the transmission dynamics of SARS-CoV-2 through the postpandemic period. *Science* 368 (6493), 860-868. doi: 10.1126/science.abb5793
- Lammers, J., Crusius, J. & Gast, A. (2020). Correcting misperceptions of exponential coronavirus growth increases support for social distancing. *Proceedings of the National Academy of Sciences of the United States of America*, 117 (28), 16264-16266. doi: 10.1073/pnas.2006048117

Appendix C

Report on “Anticipating trajectories of exponential growth”

The authors have addressed my concerns except for one. I requested that the survey instrument be included. I apologize for not being more clear in my previous review. The authors should include the survey instrument as a PDF translated into English. This is the convention in economics for English-language journals. I believe the convention is a good one as it facilitates transparency and accessibility. I believe this is a necessary final step for publication.

Appendix D

Dear Dr. Szucs, dear Dr. Parkhouse,

thank you for considering our submission “Anticipating trajectories of exponential growth” for publication. As requested by Reviewer 2 and Dr. Szucs, we now included the survey as supplementary material. Find our response to the Reviewers below.

Best regards,

Stefan Hawelka on behalf of all co-authors

Response to Reviewers.

Reviewer 1. Thank you for incorporating all the suggestions raised in the previous round. In my opinion, the manuscript is ready for publishing. Thank you for your valuable and interesting contribution.

Response. Thank you for help in improving the manuscript.

Reviewer 2. The authors have addressed my concerns except for one. I requested that the survey instrument be included. I apologize for not being more clear in my previous review. The authors should include the survey instrument as a PDF translated into English. This is the convention in economics for English-language journals. I believe the convention is a good one as it facilitates transparency and accessibility. I believe this is a necessary final step for publication.

&

Associate Editor. Please include your survey in supplementary material as recommended by R2.

Response: We now included the survey - translated to English - as a supplementary material.